# AIEgen-Based Nanomaterials for Bacterial Imaging and Antimicrobial Applications: Recent Advances and Perspectives

**DOI:** 10.3390/molecules28062863

**Published:** 2023-03-22

**Authors:** Zipeng Shen, Yinzhen Pan, Dingyuan Yan, Dong Wang, Ben Zhong Tang

**Affiliations:** 1Center for AIE Research, Shenzhen Key Laboratory of Polymer Science and Technology, Guangdong Research Center for Interfacial Engineering of Functional Materials, College of Materials Science and Engineering, Shenzhen University, Shenzhen 518060, China; 2Shenzhen Institute of Molecular Aggregate Science and Engineering, School of Science and Engineering, The Chinese University of Hong Kong, Shenzhen 518172, China

**Keywords:** aggregation-induced emission, nanomaterials, antimicrobial, multi-drug resistant

## Abstract

Microbial infections have always been a thorny problem. Multi-drug resistant (MDR) bacterial infections rendered the antibiotics commonly used in clinical treatment helpless. Nanomaterials based on aggregation-induced emission luminogens (AIEgens) recently made great progress in the fight against microbial infections. As a family of photosensitive antimicrobial materials, AIEgens enable the fluorescent tracing of microorganisms and the production of reactive oxygen (ROS) and/or heat upon light irradiation for photodynamic and photothermal treatments targeting microorganisms. The novel nanomaterials constructed by combining polymers, antibiotics, metal complexes, peptides, and other materials retain the excellent antimicrobial properties of AIEgens while giving other materials excellent properties, further enhancing the antimicrobial effect of the material. This paper reviews the research progress of AIEgen-based nanomaterials in the field of antimicrobial activity, focusing on the materials’ preparation and their related antimicrobial strategies. Finally, it concludes with an outlook on some of the problems and challenges still facing the field.

## 1. Introduction

Bacterial infections have long been a severe issue in clinical treatment [1] that can lead to serious complications (sepsis, skin disease, endocarditis, meningitis, pneumonia, etc.) and even threaten life [2,3,4]. Bacterial infections have been effectively controlled since 1928, when penicillin (the first antibiotic, followed by many others) was invented [5]. However, due to the overuse and misuse of antibiotics, antimicrobial resistance (AMR) is becoming more serious [6,7,8]. According to the World Health Organization (WHO), around 700,000 people worldwide die from multi-drug resistant (MDR) bacterial infections every year, and the number could reach 10 million by 2050 [9,10,11]. Because developing antibiotics against MDR bacterial infections is time and energy consuming, the rate of antibiotic development can hardly keep up with the rate of production of MDR bacterial infections. Therefore, there is an urgent need to develop a new alternative antibacterial strategy that not only has good broad-spectrum antibacterial activity, but also avoids the development of resistance as much as possible.

Biomedical materials developed rapidly over the last decade. Antimicrobial therapy based on nanomaterials is considered to be a promising antimicrobial strategy [12]. It mainly includes antibacterial polymers [13,14,15,16], photothermal therapy (PTT) [17,18,19], photodynamic therapy (PDT) [20,21,22,23], the specific delivery and stimulation-triggered release of antibiotics based on nanomaterials [24,25,26], catalytic killing of bacteria and anti-virulence therapy based on nanoenzymes [27], etc. In contrast to single materials, nanomaterials can be integrated with a variety of different antibacterial materials to build nanoscale diagnostic and therapeutic platforms and realize multi-pronged antimicrobial strategies for efficient and broad-spectrum antimicrobial therapy.

Antibacterial nanomaterials based on aggregation-induced emission luminogens (AIEgens) have received a lot of attention recently [28,29,30]. As a fluorescent material, AIEgens can be used for fluorescence imaging for microorganisms [31,32,33,34,35]. Existing detection methods, such as polymerase chain reaction (PCR), DNA microarrays [36], targeted specific immunoassay [37], mass spectrometry [38], and surface-enhanced Raman spectroscopy [39], are time-consuming, less accurate, and difficult to operate [40]. The use of AIEgens not only avoids the aggregation-caused quenching (ACQ) effect of traditional photosensitizers [41,42,43], but it also provides a good imaging tracer for bacteria, allows for real-time, dynamic observation of the interaction process between nanomaterials and bacteria, and reveals the antibacterial mechanisms of materials [44,45], which all benefit from AIEgens’ low background, high signal-to-noise ratio, and non-invasive real-time imaging [46,47,48]. On the other hand, AIEgens, as photosensitizers, enable photodynamic and photothermal treatment of pathogenic microorganisms. AIEgens can not only achieve long-wavelength excitation and emission through the regulation of molecular structures, but also achieve combined photodynamic and photothermal therapy for bacterial infections through the regulation of energy levels. Compared with traditional antibiotics, this antibacterial method offers good broad-spectrum antibacterial activity, and it does not easily produce drug resistance. In addition, compared with heavy metal ions, AIEgen, as an organic material, has less cytotoxicity and better biocompatibility. Phototherapy, as a non-invasive technique with high spatial and temporal accuracy, attracted wide attention in the field of antibacterials [49]. AIE-active photosensitizers can not only kill pathogens but also act as immune effector molecules to initiate and regulate the host’s immune defence system [50,51,52]. Therefore, AIEgens can effectively recognize and inhibit bacterial growth and reduce bacterial drug resistance under imaging guidance [53].

Inspired by this, we focused on recent research advances in the antimicrobial domain based on AIEgen-based nanomaterials and introduced some new antimicrobial nanomaterials and antimicrobial strategies. In this review, we focus on a number of nanotherapeutic systems constructed by AIEgens with polymers, antibiotics, metal complexes, and peptides as well as some of the novel antibacterial therapeutic strategies involved (Figure 1). Finally, it is hoped that the review of this paper will inspire more intensive research in the new frontier field of antibacterial nanomaterials.

## 2. Nanomaterials with AIEgens—Polymers for Antimicrobial Application

AIEgens were proven to have good bactericidal effects against various microorganisms in recent years [54,55,56,57]. Although small molecules alone are beneficial for photodynamic therapy of bacteria, their photostability is relatively poor compared to polymers; thus, the development of polymer-based AIEgens is highly necessary. Some commonly used methods mainly cover drugs or target molecules and the self-assembly of polymers to build a nano-assembly system to achieve multifunctional requirements. Porous, hollow, polymeric capsules and amphiphilic polymers are commonly used materials in these assembly systems [58,59]. Recently, Huang et al. [60] designed a polymer-based NIR-II AIEgen PDTPTBT, which was then encapsulated with liposomes into a nanomaterial (L-PDTPTBT) to improve the dispersion and biocompatibility of the material, as shown in Figure 1a. In vitro experiments demonstrated that L-PDTPTBT has excellent photothermal conversion effects under 808 nm laser irradiation, with the highest temperature reaching 55 °C, which can easily destroy the structure of the bacterial membrane and kill bacteria (Figure 1b,c). Subsequent in vivo experiments further revealed that L-PDTPTBT had an excellent bactericidal effect on both diabetic wound infections and subcutaneous bacterial infections, and the number of bacteria was significantly reduced (Figure 1d,e).

Polymer-based AIE molecules do have excellent photostability, but the degradation performance of the polymer is also a concern [61]. As an antibacterial drug, it is necessary not only to have good antibacterial properties, but also to minimize the biological toxicity of the drug. This requires antibiotics to degrade metabolism well in the human body and to reduce the amount of time retained in the body. Based on this, Chen et al. [62] designed a novel antimicrobial polymer with ester bonds connecting the polymeric backbone and functional segments and then cleverly introduced AIEgens into the polymer system to successfully enable the imaging and killing of bacteria (Figure 2a). The cationic amino segments of the polymer backbone can interact with the negatively charged bacterial membrane and destroy the integrity of the membrane. The lipophilic alkane segments can be inserted into the bacterial membrane, causing a distortion of the bacterial membrane structure that promotes its destruction and accelerates the death of the bacteria. The introduction of ester bonds can give antimicrobial drugs good antibacterial activity in the short term. In the biological environment, they degrade well due to the hydrolysis of lipases, thereby reducing the harm to the body. Subsequent experimental results also show that the antibacterial rate of Q-PGEDA-OP/TPE system on AMO^r^
*S. aureus* (Figure 2c) and AMO^r^
*E. coli* (Figure 2d) was more than 99%, and the subsequent SEM results show that the membrane structure of the bacteria was significantly damaged (Figure 2e).

## 3. Nanomaterials with AIEgen—Antibiotics for Antimicrobial Application

There is no doubt that antibiotics have made remarkable progress in treating bacterial infections since 1928 [63,64]. However, with the overuse of antibiotics and the abuse of broad-spectrum antibiotics, the number of infections caused by MDR bacteria has increased dramatically in recent years [65]. In addition, intracellular bacteria are one of the deadliest causes of drug resistance. After being ingested by phagocytes, the bacteria can escape from the endosomes and proliferate in the cytoplasm [66]. In addition, it is difficult for most antibiotics to play a bactericidal role due to their poor cellular penetration and short intracellular retention time [67]. Thus, how to deliver antibiotics exactly to the site of bacterial infection is important. The emergence of biodegradable nanomaterials has solved this problem well. Currently, commonly used nanomaterials such as liposomes, nanoparticles, and micelles are widely used for the intracellular delivery of antibiotics [68].

Studies have shown that macrophages infected with intracellular bacteria produce a number of different intracellular signals, such as the high expression of various enzymes by intracellular bacteria such as lipase, phosphatase, and phospholipase [69,70]. In addition, cationic polypeptides, hydrophobic carbon chains, and ferric carrier chelates were shown to enhance the ability to target bacteria [71,72]. Based on this, Chen et al. [73] designed a nanomaterial, mPET@D_Fe_C, that enables macrophage targeting, intracellular bacteria that trigger drug release, and real-time fluorescence monitoring. First, the AIE fragment was introduced into the polymeric carrier Man-g-P(EPE-r-TPE) via copolymerization, and then ciprofloxacin (CIP) and deferoxamine (DFO) were connected to obtain the siderophore–antibiotic conjugate D_Fe_C (improve the targeting of intracellular bacteria). Then, Man-g-P(EPE-r-TPE) and D_Fe_C were self-assembled to obtain the final nanoparticles of mPET@D_Fe_C. Because there is a fluorescence resonance energy transfer (FRET) effect between the polymer carrier containing the AIE fragment and the drug iron carrier conjugate, the drug release process can be monitored in real time through changes in the fluorescence emissions of the nanoparticles (Figure 3a). Second, mPET@D_Fe_C can effectively enter macrophages through mannose-mediated endocytosis and then achieve polymer degradation and D_Fe_C release under the action of lipase and phospholipase secreted by intracellular bacteria with strong specificity and no damage to normal macrophages (Figure 3b). Finally, when D_Fe_C is ingested by bacteria to realize sterilization, the FRET effect is stopped and the AIE effect is restored, thereby realizing the fluorescence monitoring of intracellular bacteria (Figure 3b).

In addition to loading drugs through the self-assembly of organic macromolecules, antibiotics can also be loaded with the help of some existing carriers. Organic silica nanoparticles can not only improve the drug loading rate, but also improve the biocompatibility of materials by selecting suitable silica precursors, which are good drug carriers [13,74,75,76]. Inspired by the guiding role of surfactants in the preparation of nanoparticles, Yan et al. [77] used the AIE molecule MeOTTVP as the framework and adopted the two-template assisted one-pot method to prepare organic silica nanoparticles (AIE-ONs). This nanoparticle can be added to doxorubicin (DOX) to realize the diagnosis and treatment of cancer. Moreover, the antibiotic rifampicin (RF) can be loaded to effectively treat bacterial infections (Figure 4a). Subsequent antibacterial experiments further proved that AIE-ONs can use the MeOTTVP fluorescence of the AIE molecule to achieve targeted imaging of *S. aureus* and *E. coli* (Figure 4b). In addition, under the irradiation of white light, ROS produced by AIE molecules and the loaded drug rifampicin can kill bacteria well, with a bactericidal rate of up to 99.9% (Figure 4c).

In conclusion, nanomaterials constructed with AIE molecules and antibiotics can greatly improve the therapeutic effects against bacterial infections. It not only avoids the excessive use of antibiotics leading to bacterial resistance, but it also further improves the antibacterial effect of AIE molecules alone, which is a promising therapeutic strategy.

## 4. Nanomaterials with AIEgen—Peptides for Antimicrobial Application

Antimicrobial peptides are considered to be ideal antimicrobial agents due to their excellent broad-spectrum antibacterial activity and low drug resistance [51,78,79,80]. Although antimicrobial peptides have good antibacterial effects against both gram-positive and gram-negative bacteria, little is known about their bactericidal mechanisms [81,82,83]. Therefore, realizing a way to dynamically monitor the interaction between antimicrobial peptides and bacteria is very important [84]. Traditional imaging techniques such as SEM and TEM can only achieve static observation, not dynamic monitoring [85,86]. On the contrary, fluorescence signal monitoring can realize not only dynamic observation but also continuous monitoring [87,88]. The introduction of AIE fluorescent probes can effectively solve this problem. Compared with the fluorescence quenching caused by the aggregation of traditional fluorescent probes, the fluorescence signal of AIE fluorescent probes are significantly enhanced during aggregation, which can allow for good tracer imaging on the target [43].

Based on this, Chen et al. [89] introduced AIE fluorescent probe HBT into antimicrobial peptide HHC36 to obtain the AIE active nanomaterial AMP-2HBT (Figure 5a). Subsequent antibacterial experiments showed that the introduction of AIE fluorescent probes did not damage the antibacterial activity of the antimicrobial peptides, and the growth of *E. coli* was well-inhibited at the concentration of 20 μM (Figure 5b). In addition, as displayed in Figure 5c, there was a strong green fluorescence signal on the surface of the bacterial membrane, indicating that the antimicrobial peptides mainly adhered to the surface of the bacteria and caused the internal nucleic acid or protein to leak by destroying the structure of the bacterial membranes, thereby leading to the death of the bacteria. In conclusion, the combination of AIE molecules and antimicrobial peptides not only has a good bactericidal effect, but the process can also be dynamically monitored in real time.

Compared to external bacterial infections, the treatment of intracellular bacterial infections has always been a difficult problem. The combination of AIE molecules and peptides can not only realize the imaging and sterilization of extracellular bacteria mentioned previously, but it can also have a good effect on the imaging and treatment of intracellular bacterial infections. Qi et al. [90] designed a nano-probe, PyTPE-CRP, specific to casp-1 that can be used for the imaging and treatment of intracellular bacterial infections by taking advantage of the property that macrophages can recognize bacterial infections and induce the activation of casp-1, as shown in Figure 6a. PyTPE-CRP is composed of two parts. As a reactive part, CRP can cleave between amino acids Asp and Ala (red dotted line in Figure 6a) during bacterial infections through the activation of casp-1 enzymes. The resulting PyTPE-CRP residues spontaneously self-assemble into aggregates and accumulate on macrophages containing bacteria. To achieve the specific imaging of bacteria-infected macrophages, AIE fluorophore (PyTPE), in its molecular state, almost does not emit light, but as an aggregate, it shows strong emission, which can realize the imaging and killing of intracellular bacteria. As can be seen in Figure 6b, the fluorescent probes after lysis were evenly distributed around the intracellular bacteria. Under the irradiation of white light, the fluorescent probe after cracking produced a large number of ROS around the bacteria, achieving photodynamic therapy against the bacteria (Figure 6c). At the concentration of 20 μM, it had a good antibacterial effect on *S. aureus*, and the minimum inhibitory concentration was as low as 15 μM (Figure 6d,e).

Most nanomaterials containing antimicrobial peptides kill bacteria directly, mainly by producing ROS, heat, or by disrupting the structure of the bacterial membrane. Recently, a new antimicrobial peptide, HD6, attracted attention [91,92,93]. Unlike traditional antimicrobial peptides, HD6 traps microbial pathogens through a network of self-assembled fibres, thereby preventing their invasion [94]. Based on this, Fan et al. [95] constructed a bionic analogue peptide, HDMP, which can simulate natural HD6 and achieve the self-assembly process induced by the ligand–receptor interaction to achieve the purpose of bacterial recognition and capture, as shown in Figure 7a. HDMP mainly consists of three parts (Figure 7b). The bacterial targeting recognition part, RLYLRIGRR, can bind to the unique component LTA in Gram-positive bacteria and has specific targeting ability. The peptide skeleton, KLVFF, which mimics the fibre structure in HD6, forms a network of fibres that capture bacteria. The last part is BP, which not only promotes the self-assembly of peptides into nanoparticles to improve the ability of intravenous drug delivery, but it also has the AIE effect, which can monitor the distribution of antimicrobial peptides via fluorescence in real time. From the perspective of antibacterial mechanisms, it first self-assembles into nanoparticles in vitro and then binds specifically to bacterial walls, then transforming into a fibre network, triggered by ligand–receptor interaction, to achieve the purpose of capturing and inhibiting bacterial invasion. As can be seen in the SEM photo in Figure 7c, compared with the blank control group, there were many fibre network adhesions on the surface of bacteria in the experimental group, indicating that the HDMP nanomaterials successfully realized the recognition and capture of bacteria. Further in vivo infection experiments in mice also demonstrated that HDMP nanomaterials had a good therapeutic effect on *S. aureus* abscesses and bacteraemia.

In conclusion, the nanomaterials formed by antimicrobial peptides and AIE molecules not only improve the antibacterial effect of the materials, but also do not easily produce drug resistance. More importantly, the introduction of AIE molecules can give nanomaterials the function of real-time dynamic monitoring, providing a powerful tool for further understanding the antibacterial mechanisms of nanomaterials.

## 5. Nanomaterials with AIE Metal Complexes for Antimicrobial Applications

Metal-organic frameworks (MOFs) are widely used in drug delivery because of their excellent loading capacity, easy removal, and low biotoxicity [92,96]. More importantly, their adjustable chemistry gives MOFs multiple stimulation–response drug release properties (pH, magnetic field, ions, temperature, and light) to adapt to different physiological environments [96,97].

Based on this, Mao et al. [98] developed a strategy for bacterial detection and treatment by combining MOFs with metabolic labelling technology. First, MIL-100 (Fe) was selected as the carrier of the metabolic marker molecule 3-azide-d-alanine (_D_-AzAla), which then self-assembled with F-127 to obtain the composite nanomaterial D-AzAla@MIL-100, as shown in Figure 8a. After intravenous injection of _D_-AzAla@MIL-100, the nanomaterial can be enriched at the site of bacterial infection through the EPR effect. Subsequently, under the action of H_2_O_2_ secreted by immune cells, MIL-100 (Fe) dissociates and releases _D_-AzAla, which can then be specifically absorbed by bacteria in the infected area. During this process, unnatural azide groups are be expressed on the bacterial wall to achieve bacterial labelling (Figure 8b). Subsequently, selective fluorescent labelling and precise sterilization were achieved via biological orthogonal reactions and PDT with the AIE photosensitizer TPETM (Figure 8c). As shown in Figure 8e, compared to the blank group, mice in the D-AzAla@MIL-100 pretreatment group showed stronger fluorescence images, a longer half-life, and a significantly reduced number of bacteria infected at the wound site (Figure 8e,f).

In addition, gold nanomaterials are widely used in the biomedical field because of their excellent biocompatibility [99,100,101]. Notably, when the size of the gold nanomaterials is reduced to the subnanometer scale, these ultra-small Au NCs begin to take on unique physicochemical and biological properties [102]. These properties make Au NCs an excellent candidate for combination therapy with other antimicrobial agents, such as AMPs [103]. Zheng et al. [104] combined the antimicrobial gold nanocluster AuDAMP with the antimicrobial peptide daptomycin Dap to produce a new antimicrobial nanocomplex (Figure 9). The conjugation of gold nanoclusters with AMPs in this compound results in AIE enhancement. The antibacterial mode of Dap mainly occurs through the lipophilic tail inserted into the bacterial cell membrane with the aid of calcium, causing rapid cell membrane damage and potassium ion efflux. Membrane damage encourages the introduction of antimicrobial compounds into the bacteria and leads to more severe bacterial damage at the subcellular level. This strategy provides a new perspective for the synthesis of novel antimicrobial agents and AIE-type fluorescent materials, and it provides a way to further study the specific mechanisms behind the conjugation-induced AIE effect.

## 6. Other Nanomaterials with AIEgen for Antimicrobial Applications

On one hand, related diseases caused by bacterial infection seriously threaten human health [98,105,106,107]; on the other hand, some bacterial communities, such as gut microbes, are essential to human health [91,108]. Therefore, we have pursued improvement of the targeting and killing efficiency of antibacterial agents. Studies have shown that phages are host-specific and can evolve synchronously to infect MDR bacteria [109,110,111]. However, bacteriophages alone have low antibacterial efficiency, and it is ineffective against acute infections and other severe infectious diseases [112]. In addition, due to the lack of imaging fragments, the target identification, binding, infection, and other processes of phagocytic therapy are not easy to monitor, and it is difficult to evaluate their therapeutic effects in real time.

Therefore, He et al. [113] proposed a novel strategy to bind AIEgens to phages to form a new class of antimicrobial bioconjugates (TVP−PAP) that are used to image and kill specific species of bacteria, as shown in Figure 10a. Not only does this new antimicrobial material retain the specificity of bacteriophage targeting, but the inherent fluorescence of the introduced AIEgens (TVP) also allows real-time monitoring of bacteriophage interactions. At the same time, the highly efficient photodynamic inactivation of TVP and the excellent bacteria-targeting ability of PAP synergistically endow TVP-PAP with excellent bactericidal effects, significantly exceeding the antibacterial effects of the two components individually. As can be seen in Figure 10b, TVP-PAP staining for 30 min can bind well to *P. aeruginosa* (host bacteria) to produce bright fluorescence, and staining efficiency is as high as 100%. However, non-host bacteria *A. baumannii* did not stain all of them, indicating that TVP-PAP can target bacteria accurately. Subsequent antibacterial experiments further proved that TVP-PAP not only has a good bacteria-targeting ability, but it also has a good bactericidal effect on host bacteria, with a selective bactericidal efficiency of up to 90% (Figure 10c,d).

As an inorganic nanomaterial, montmorillonite has been widely used in biomedical fields, such as intestinal diseases, drug delivery, additive manufacturing, and so on [114,115,116,117,118]. Because of its highly ordered lattice arrangement, it has a high cation exchange capacity and surface area; thus, it is a good drug transport carrier [104,119]. In addition, studies show that MMT can absorb bacteria and bacterial enterotoxins well in the body. However, in vitro, its antibacterial effect is very weak [120,121], and it is difficult to meet the needs of clinical external infection treatment with it.

Therefore, Zhang et al. [81] developed an ultra-efficient photodynamic/chemokinetic treatment platform by inserting the aggregation-induced emission (AIE) photosensitizer TPCI into nanolayers of iron-containing montmorillonite (MMT). Here, the introduction of iron atoms can achieve chemodynamic therapy and enhance the effect of photodynamic therapy. Studies show that the site of bacterial infection has a microenvironment with a low pH value and a relatively high endogenous hydrogen peroxide level. Therefore, iron ions can convert the endogenous hydrogen peroxide with low activity into highly toxic hydroxyl radicals under weak acid through the Fenton reaction, thereby inducing bacterial inactivation. Therefore, the TPCI/MMT treatment system can not only carry out efficient PDT through the production of singlet oxygen, but it can also continuously implement CDT by converting endogenous H_2_O_2_ into highly toxic hydroxyl radicals, as shown in Figure 11a. The generation of hydroxyl radicals and singlet oxygen was subsequently verified with ESR spectroscopy (Figure 11b). The bactericidal effect of TPCI/MMT on *E. coli* and *S. aureus* under white light was more than 99% compared to the blank control group and the group receiving MMT alone (Figure 11c). In addition, it can be seen in the SEM results that TPCI/MMT mainly kills bacteria by destroying the integrity of the bacterial membrane structure (Figure 11d). Subsequently, the antibacterial effect of TPCI/MMT was evaluated with an in vivo infected wound healing assay (Figure 11e). The results showed that TPCI/MMT could effectively promote the healing of infected wounds and significantly reduce the number of bacteria in infected tissues with good therapeutic effects (Figure 11g,h).

At present, the regulation of the antibacterial properties of AIE molecules is mainly realized based on the reasonable design of the positively charged molecular framework of AIEgen and specific recognition groups [21,90,122,123]. However, due to their limited molecular skeleton, the further improvement of the antimicrobial properties of AIE molecules is greatly restricted [42]. Here, Guo et al. [124] used AIEgen DTPM as the inner core and prepared a series of AIE nanofibers that could precisely regulate the antibacterial activity by reasonably designing peptides as the recognition system. The preparation process is shown in Figure 12a. First, AIE molecules are coated with amphiphilic molecules to improve the biocompatibility of the materials. Then, the designed peptides are introduced on the surface of AIE molecules via a maleimide–mercaptan addition reaction to regulate the antibacterial activity of nanomaterials. Through mechanism analysis, it was found that this effect can be attributed to the combined action of ROS and antimicrobial peptides produced by AIE molecules, which had obvious synergistic antibacterial effects (Figure 12b). The antimicrobial activity of the materials can be precisely regulated by the modification of different antimicrobial peptides. It can be seen in Figure 12c that K18-modified nanofibers had the best bacterial adsorption effect, followed by K14. In addition, from the bactericidal effect, NFs-K18 also had a very good antibacterial effect (Figure 12d).

## 7. Summary and Perspective

AIEgen-based nanomaterials further enhance the antimicrobial effectiveness of AIEgens with the introduction of other materials and the construction of nanoplatforms while retaining the advantages of AIEgens. In this review, novel antimicrobial nanomaterials constructed from AIEgens with polymers, antibiotics, metal elements, peptides, and some other materials in recent years are presented along with some of the antimicrobial strategies involved. Finally, the types, sterilization methods, mechanisms, and application scenarios of nanomaterials in different systems are summarized in Table 1.

Although these novel antimicrobial nanomaterials have made good progress in the field of antimicrobial activity, there are still some challenges and issues to be further investigated. One such issue is finding out how to further simplify the functionalised modification steps of AIEgens and how to enhance the diversity of functionalised modifications to meet the needs of a wider range of environments. A second challenge is to discover how to make greater use of the advantages of AIEgen fluorescence imaging, how to apply it to the enhancement of antibacterial performance, and how to reflect its superiority to traditional antibiotics. In addition, there is the issue of how to resolve the difficulty in reaching infected areas via laser in deep tissue phototherapy. Finally, the biocompatibility of various nanomaterials in clinical treatment and the issues related to in vivo retention and metabolism need further validation. It is hoped that this review will inspire some inspiration for researchers to do more excellent work and advance further research in this field.

## Data Availability

Not applicable.

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
