# Peer review of "AIEgen-Based Nanomaterials for Bacterial Imaging and Antimicrobial Applications: Recent Advances and Perspectives"

_molecules, 2023, doi:10.3390/molecules28062863_

Round 1
Reviewer 1 Report
The authors aim at providing an update on recent advances and perspectives of AIEgen nanomaterials. Unfortunately, they didn’t accomplished the task. The review sounds a simple list of systems containing AIEgen and no comparison is reported. In addition, even though the authors claim that these compounds are endowed with antimicrobial activity, in all reviewed systems the biocide action is exerted by other moiety than AIEgen Thus, it appears that the key role of AIEgen is related to monitoring infection therapy rather than its treatment.
On the whole, the review doesn’t meet the quality standard of Molecules
Author Response
Thanks for the referee’s kind suggestions. The following is our response to the comments.
In this review, we mainly summarize the recent work on AIEgens-based antibacterial nanomaterials, including nanomaterials with AIEgens-polymers, nanomaterials with AIEgen-antibiotics, nanomaterials with AIEgen-peptides, nanomaterials with AIE metal complex, and other nanomaterials with AIEgen. Different nanomaterial systems possess different antibacterial modes, mechanisms and application scenarios. To make a clear comparison for each system, we add Table 1 in the revised manuscript, through which the readers can clearly see the antibacterial modes, mechanisms and application scenarios involved in each material in different systems.
The role of AIEgen molecules in nanomaterials can be divided into two main categories: imaging and therapy. The literature cited in this paper not only report the function of AIEgnes as fluorescent material to image and trace bacteria, but also the function of AIEgens as photosensitizers to generate ROS or heat under light to realize PDT and PTT towards bacteria. For example, AIEgens in Figure 5 is mainly used in the material to dynamically monitor the contact process between antibacterial peptides and bacteria, and study their antibacterial mechanism. In Figure 6, AIEgen produces ROS in nanomaterials to kill bacteria mainly through light activation. AIEgens shown in Figure 1 can not only realize fluorescence imaging in IR-II region, but also kill bacteria by ablating them with heat generated by light.
According to the reviewer's suggestion, we have carefully modified the manuscript. We sincerely hope that the reviewer will be satisfied with our revised manuscript.
Reviewer 2 Report
This review concluded recent advance in the AIEgens based nanomaterials for antimicrobial. The AIEgens could enable fluorescent tracing of microorganisms and production of the ROS.
In this review, the authors focus on a number of nanotherapeutic systems constructed by AIEgens with polymers, antibiotics, metal complexes, peptides as some other antibacterial therapeutic strategies. Finally, the conclusion and perspectives of AIEgens based materials are summarized. Overall, this is a well-organized manuscript, and some suggestions should be addressed before publication.
1. A summary picture should be outlined at the beginning of the article.
2. The author needs to emphasize exactly what advantages AIEgens-based nanomaterials has over other materials.
3. Some figure signs should be seriously revised, for instance, Figure 5c also have the A-F inset.
4. What is the most important antibacterial mechanism of AIEgens-based materials? Especially for the drug-resistant bacteria.
5. Whether these antimicrobial groups will have destructive effects on other normal cells?
6. A summary comparative table should be added.
7. For comparing the antibacterial effects, some related research about the antibacterial polymers should be cited to highlight the potential applications of these materials. Biomacromolecules 2020, 22 (2), 732-742, Materials Today Chemistry, 2022, 26: 101252.
Author Response
Thanks very much for the referee’s constructive comments! The following is the point-by-point response to the comments involving the other issues mentioned by the referee.
Concerns of main text:
1. A summary picture should be outlined at the beginning of the article.
Reply:
Thanks for the referee’s kind suggestion. We have added Scheme 1 in the introduction, which provides a good overview of the paper and shows the differences and connections between the different antibacterial nanomaterials.
The author needs to emphasize exactly what advantages AIEgens-based nanomaterials has over other materials.
Reply:
Thanks a lots for the referee’s constructive comments! The advantages of AIEgens-based nanomaterials over other materials are mentioned in the third paragraph of the first part "Introduction". The advantages can be divided into the following aspects: First, as a fluorescent material, AIEgens significantly enhances fluorescence in the aggregation state compared with traditional fluorescent materials (aggregation leads to fluorescence quenching), which is very conducive to bacterial imaging. In addition, it has the advantages of low signal-to-noise ratio, strong photobleaching resistance and adjustable emission. The introduction of AIEgens not only allows us to observe the morphology of bacteria through fluorescence signals, but also allows us to monitor the contact process between materials and bacteria in real time to study the antibacterial mechanism. Second, as a photosensitizer, AIEgens can not only achieve long-wavelength excitation and emission through the regulation of molecular structure, but also achieve the combined photodynamic and photothermal therapy of bacteria through the regulation of energy levels. Compared with traditional antibiotics, this antibacterial method not only has good broad-spectrum antibacterial activity, but also is not easy to produce drug resistance. In addition, compared with heavy metal ions, AIEgens, as an organic material, has less cytotoxicity and better biocompatibility. Based on this, in the third section of the first part, "Introduction", the advantages of AIEgens-based nanomaterials compared with other materials are also supplemented correspondingly.
- Some figure signs should be seriously revised, for instance, Figure 5c also have the A-F inset.
Reply:
Thanks to the referee for his kind advice. We have carefully modified the pictures. In addition, we have also checked other pictures in the article and adjusted the pictures (Figure 1b, Figure 2e) with similar problems or other problems.
- What is the most important antibacterial mechanism of AIEgens-based materials? Especially for the drug-resistant bacteria.
Reply:
Thanks very much for the referee’s constructive comments! The antibacterial mechanism of AIEgens-based materials is also summarized in the newly added Table 1, which can be roughly divided into the following categories. The first type, photothermal antibacterial, mainly through the heat generated by the light material causes damage to the bacterial membrane, K+ leakage leads to bacterial death. The second type is photodynamic antibacterial. ROS (1O2, ·OH) is produced by the material through light, which damages the structure of the bacterial membrane and causes bacterial death due to the leakage of bacterial contents. The third type is the electrostatic interaction of positive and negative charges. Since the surface of the bacterial membrane is mainly peptidoglycan, which presents a net negative charge on the surface of the membrane, the positively charged material can have electrostatic reaction with the bacteria, affecting the normal physiological function of the bacteria and causing the instability of the bacterial structure, thus inducing the death of the bacteria. The fourth type, antibiotics antibacterial, this way of sterilization is mainly dependent on the bactericidal activity of antibiotics themselves, but due to the production of bacterial resistance, so it is often used together with other antibacterial materials to play a synergistic antibacterial effect. Due to the multi-component and multifunctional properties of nanomaterials, the above four antibacterial methods are often used in combination with multiple antibacterial methods. The same is true for the bactericidal mechanisms of drug-resistant bacteria, because these bactericidal mechanisms, unlike antibiotics, are not easy to develop resistance to bacteria and have broad-spectrum antibacterial activity.
Whether these antimicrobial groups will have destructive effects on other normal cells?
Reply:
First of all, thank you for your kind suggestions. As to whether antibacterial groups will have destructive effects on normal cells, my answer is that they will have certain toxicity to normal cells, but this toxicity meets the requirements of biological safety, and we will prove it through a series of experiments, such as cytotoxicity evaluation, cell death staining, hemolysis experiment, and so on. In the example of Figure 1 (Biomaterials 2022, 286, 121579), the author evaluated the biological safety of materials through CCK-8 method and hemolysis experiment. The results showed that under the effective bactericidal concentration (200 µg mL-1), the viability of normal cells remained about 95% (less than 80% considered to be cytotoxic), and no obvious hemolysis occurred, indicating that the material had very little toxicity to normal cells and had excellent biocompatibility. In addition, from another perspective, some biocompatible materials will also be introduced in the process of preparing AIEgen molecules into nanomaterials. The addition of these materials will also greatly reduce the toxicity of the materials to normal cells and improve the biosafety performance of the materials. That's the answer to the question of whether antimicrobial groups can cause damage to other normal cells.
A summary comparative table should be added.
Reply:
Thank you for your suggestion, which is a very good suggestion. Based on this, we add Table 1 at the end of the paper, which mainly summarizes antibacterial nanomaterials under different systems based on AIEgen. We compares and analyzes the types of materials, antibacterial modes and mechanisms, application scenarios for better understanding the development of related fields to readers.
For comparing the antibacterial effects, some related research about the antibacterial polymers should be cited to highlight the potential applications of these materials. Biomacromolecules 2020, 22 (2), 732-742, Materials Today Chemistry, 2022, 26: 101252.
Reply:
Thank you for your suggestion. The relevant literature you mentioned have been cited in the revied manuscript.